# Evaluation of the Preclinical Efficacy of Lurbinectedin in Malignant Pleural Mesothelioma

**DOI:** 10.3390/cancers13102332

**Published:** 2021-05-12

**Authors:** Dario P. Anobile, Paolo Bironzo, Francesca Picca, Marcello F. Lingua, Deborah Morena, Luisella Righi, Francesca Napoli, Mauro G. Papotti, Alessandra Pittaro, Federica Di Nicolantonio, Chiara Gigliotti, Federico Bussolino, Valentina Comunanza, Francesco Guerrera, Alberto Sandri, Francesco Leo, Roberta Libener, Pablo Aviles, Silvia Novello, Riccardo Taulli, Giorgio V. Scagliotti, Chiara Riganti

**Affiliations:** 1Department of Oncology, University of Torino, 10043 Orbassano, Italy; dario.anobile@edu.unito.it (D.P.A.); paolo.bironzo@unito.it (P.B.); francesca.picca@unito.it (F.P.); deborah.morena@unito.it (D.M.); luisella.righi@unito.it (L.R.); francesca.napoli@unito.it (F.N.); mauro.papotti@unito.it (M.G.P.); apittaro@cittadellasalute.to.it (A.P.); federica.dinicolantonio@unito.it (F.D.N.); chiara.gigliotti@unito.it (C.G.); federico.bussolino@unito.it (F.B.); valentina.comunanza@unito.it (V.C.); francesco.leo@unito.it (F.L.); silvia.novello@unito.it (S.N.); riccardo.taulli@unito.it (R.T.); 2Thoracic Unit and Medical Oncology Division, Department of Oncology at San Luigi Hospital, University of Torino, 10043 Orbassano, Italy; 3Center for Experimental Research and Medical Studies (CeRMS), City of Health and Science University Hospital di Torino, University of Torino, 10126 Torino, Italy; 4Department of Medical Sciences, University of Torino, 10126 Torino, Italy; marcello.lingua@edu.unito.it; 5Pathology Unit, San Luigi Hospital, University of Torino, 10043 Orbassano, Italy; 6Pathology Unit, City of Health and Science University Hospital, 10126 Torino, Italy; 7Interdepartmental Centre for Studies on Asbestos and Other Toxic Particulates, University of Torino, 10125 Torino, Italy; 8Candiolo Cancer Institute—FPO, IRCCS, 10060 Candiolo, Italy; 9Department of Surgical Science, University of Torino, 10126 Torino, Italy; francesco.guerrera@unito.it; 10Department of Thoracic Surgery, City of Health and Science University Hospital, 10126 Torino, Italy; 11Thoracic Surgery Division, San Luigi Hospital, University of Torino, 10043 Orbassano, Italy; al.sandri@sanluigi.piemonte.it; 12Department of Integrated Activities Research and Innovation, Azienda Ospedaliera SS. Antonio e Biagio e Cesare Arrigo, 15121 Alessandria, Italy; rlibener@ospedale.al.it; 13Research and Development Department, PharmaMar, 28770 Colmenar Viejo, Madrid, Spain; paviles@pharmamar.com; 14Interdepartmental Research Center of Molecular Biotechnology, University of Torino, 10126 Torino, Italy

**Keywords:** MPM, lurbinectedin, DNA damage response

## Abstract

**Simple Summary:**

The marine drug lurbinectedin revealed an unprecedented efficacy against patient-derived malignant pleural mesothelioma cells, regardless of the histological type and the BAP1 mutation status. By inducing strong DNA damages, it dramatically arrested cell cycle progression and induced apoptosis. These results may be translated into the use of lurbinectedin as an effective agent for malignant pleural mesothelioma patients.

**Abstract:**

Background: Malignant pleural mesothelioma (MPM) is a highly aggressive cancer generally diagnosed at an advanced stage and characterized by a poor prognosis. The absence of alterations in druggable kinases, together with an immune-suppressive tumor microenvironment, limits the use of molecular targeted therapies, making the treatment of MPM particularly challenging. Here we investigated the in vitro susceptibility of MPM to lurbinectedin (PM01183), a marine-derived drug that recently received accelerated approval by the FDA for the treatment of patients with metastatic small cell lung cancer with disease progression on or after platinum-based chemotherapy. Methods: A panel of primary MPM cultures, resembling the three major MPM histological subtypes (epithelioid, sarcomatoid, and biphasic), was characterized in terms of BAP1 status and histological markers. Subsequently, we explored the effects of lurbinectedin at nanomolar concentration on cell cycle, cell viability, DNA damage, genotoxic stress response, and proliferation. Results: Stabilized MPM cultures exhibited high sensitivity to lurbinectedin independently from the BAP1 mutational status and histological classification. Specifically, we observed that lurbinectedin rapidly promoted a cell cycle arrest in the S-phase and the activation of the DNA damage response, two conditions that invariably resulted in an irreversible DNA fragmentation, together with strong apoptotic cell death. Moreover, the analysis of long-term treatment indicated that lurbinectedin severely impacts MPM transforming abilities in vitro. Conclusion: Overall, our data provide evidence that lurbinectedin exerts a potent antitumoral activity on primary MPM cells, independently from both the histological subtype and BAP1 alteration, suggesting its potential activity in the treatment of MPM patients.

## 1. Introduction

Malignant pleural mesothelioma (MPM) is a rare but extremely aggressive type of cancer arising from pleural mesothelium and is highly associated with asbestos exposure. The disease is characterized by a long latency between initial exposure to asbestos and the clinical onset of the disease (30–50 years) and, although in Western regions the peak was expected in the 2020s [1], the ongoing use of asbestos in developing countries could lead to a persistence of new cases in the next decades [2]. MPM is classified into three major histological subtypes: epithelioid, sarcomatoid, and biphasic. While the epithelioid subtype occurs more frequently, accounting for approximately 60% of cases, and correlates with a better outcome, the sarcomatoid subgroup represents 10–20% of the cases and is characterized by a worse prognosis [3,4]. Independently from the morphology, the MPM tumor microenvironment is particularly enriched of immunosuppressive cells, which makes this tumor particularly refractory to different therapies [5,6,7,8,9,10]. Moreover, MPM is generally diagnosed in advanced stage, minimizing the role of curative treatments. For advanced-stage disease, the first-line systemic treatment consists of cisplatin and pemetrexed [11], a combination that prolongs the median survival time of only 3 months. Recently, the combination of immune checkpoint inhibitors directed against programmed death-1 (PD-1) and cytotoxic-T-lymphocyte-associated protein 4 (CTLA-4) showed its superiority over chemotherapy in previously untreated and unresectable MPM, especially in non-epithelioid tumors [12]. Conversely, no second-line standard therapy has been approved, despite the pre-clinical and the clinical evaluation of different therapeutic agents [13,14].

The genomic landscape of MPM reveals a low mutational burden with inactivating alterations mainly on oncosuppressors (BAP1, CDKN2A, NF2, TP53, LATS2, and SETD2) [15,16,17,18] thus precluding the use of molecular therapies against activated oncogenes. Among the oncosuppressors, BAP1 (BRCA1-associated protein) alterations account from 30% to 60% of cases [15,17,19,20]. Indeed, BAP1 germline mutations are known to predispose to mesothelioma and other cancer-associated syndromes [21,22] thus indicating a critical role for this deubiquitinase in suppressing tumor development. BAP1 regulates different biological processes among which chromatin modification, cell cycle, apoptosis, ferroptosis, cell metabolism, and differentiation [23]. Notably, BAP1 is involved in DNA synthesis, DNA duplication under stress conditions [24,25], and DNA damage response, by modulating the function of the BRCA1/BARD1 (BRCA1 Associated RING Domain 1) complex and coordinating the recruitment of RAD51 to the damaged DNA loci [26,27].

Lurbinectedin (PM01183) is a marine-derived anticancer drug that exerts a potent antitumor activity in different cancer cell lines and xenografts models and is currently under clinical evaluation in several tumor types [28,29,30,31,32,33,34,35]. Recently, the FDA has released a conditional approval for lurbinectedin for the treatment of second-line metastatic small cell lung cancer patients [36] while promising antitumor activity has been reported in MPM patients in second- and third-line [37]. However, there are no data available on the role of lurbinectedin as monotherapy or in combination in the first-line treatment of MPM. At the molecular level, lurbinectedin covalently binds CG-rich sequences in the DNA minor groove. The presence of the drug on the DNA helix inhibits the transcriptional process and is associated with the generation of DNA breaks [28]. Moreover, the interaction of lurbinectedin with both DNA strand breaks also interferes with the enzymes involved in the DNA damage response [38].

Here, we report about the potential efficacy of lurbinectedin in a panel of primary MPM cultures. Specifically, we demonstrated that lurbinectedin is strongly effective at nanomolar concentration and interferes with the transforming properties of MPM in a way that is independent of the BAP1 status and histological classification. With the caveat that our cell cultures were derived from diagnostic biopsies or surgical resections, our data indicate that lurbinectedin could potentially be explored in the management of patients with advanced MPM as second-line treatment or part of combination treatment in first-line.

## 2. Results

### 2.1. Primary Mesothelioma Cell Cultures Characterization

Twelve primary MPM cell lines, derived from patients with different histology, were stabilized as 2D cultures (Figure 1A). Flow cytometry for pan-cytokeratin (Figure 1B), immunohistochemical analysis (Figure 1C and Table 1), and immunoblotting for the BAP1 status (Figure 1D) were used to characterize the MPM cell lines. Notably, our panel (6 BAP1+ and 6 BAP1− cultures) was representative of the three major MPM histological subtypes (epithelioid, sarcomatoid, and biphasic) (Appendix A).

### 2.2. Lurbinectedin Exerts Anti-Proliferative Effects in Patient-Derived Mesothelioma Cells

As shown in Figure 2, lurbinectedin decreased the viability of MPM cells in a dose-dependent manner, with an IC_50_ in the low nanomolar range for all cell lines (Table 2), independently from the BAP1 status and the histological subtype (Figure 2A–D). Indeed, although the IC_50_ was slightly higher in BAP1− vs. BAP1+ cells (Figure 2C) as well as in the sarcomatoid/biphasic vs epithelioid histotype (Figure 2D), the difference was not statistically significant. Notably, UPN6, UPN10, and UPN12 received trabectedin as second-line treatment and their overall survival was <12 months (Appendix A). The cell lines derived from these patients had indeed the highest IC_50_ in the panel analyzed, but it was below 5 nM for all of them (Table 2).

### 2.3. Long-Term Lurbinectedin Treatment Impacts on MPM Transforming Abilities

Since mesothelioma is particularly resistant to conventional chemotherapy, we evaluated the long-term effect of lurbinectedin in terms of inhibiting cell proliferation by performing a crystal violet viability assay. Also in this setting, nanomolar concentrations of lurbinectedin dramatically reduced cell growth (Figure 3A,B). Furthermore, we extended our analysis by testing lurbinectedin ability to interfere with the anchorage-independent growth of MPM cells. The number of visible colonies was markedly decreased upon treatment, showing long-term anticancer efficacy (Figure 3C,D). Importantly, the consistent reduction in anchorage-independent growth showed no differences between BAP1+ and BAP1− cells, suggesting that lurbinectedin strongly impairs the tumorigenic potential of MPM cells, independently from the BAP1 status.

### 2.4. Lurbinectedin Treatment Interferes with Cell Cycle Progression

To study the molecular basis of this anti-proliferative activity, we analyzed the effect of lurbinectedin on cell cycle regulation. While we observed variable changes in the percentage of cells in the G2/M-phase, indicating an unlikely strong mitotic arrest, we observed a constant accumulation of cells in the S-phase (Figure 4 and Appendix A). This event occurred in both BAP1+ and BAP1− cells, suggesting that lurbinectedin-mediated perturbation of the cell cycle is BAP1-independent.

### 2.5. Lurbinectedin Induces a Profound DNA Damage Coupled with Strong Apoptosis

Among the pleiotropic mechanisms of action of lurbinectedin [28,38] the increase of S-phase arrested cells is suggestive of irreversible DNA damage. Indeed, lurbinectedin induced a significant increase in round-shaped and dense cells (Appendix A). The presence of irreversible DNA fragmentation was evaluated by the Single Cell Gel Electrophoresis (SCGE). Specifically, in both BAP1+ and BAP1− cells lurbinectedin induced a dose-dependent genomic fragmentation (Figure 5A,B). The presence of genotoxic stress was confirmed by the increase in the phospho (Ser345) Chk1 and phospho (Thr68) Chk2 (Figure 5C,D), two cell cycle checkpoints that block DNA replication after being phosphorylated by the DNA-damaging sensors ATM/ATR kinases [39]. Moreover, in lurbinectedin-treated cells, we observed the accumulation of phospho (Ser15) p53 and phospho (Ser139) H2AX (Figure 5C,D), two additional targets of ATM/ATR kinases that are generally phosphorylated in response to DNA strand breaks and stalled replication [40,41]. This provided additional evidence of the strong DNA damage induced by lurbinectedin, which is also responsible for cell growth arrest (Figure 4 and Appendix A). Such mitotic catastrophe is often coupled with apoptosis [40]. Accordingly, lurbinectedin treatment resulted in a strong induction of apoptosis (Figure 6A,B) as also shown by the dose-dependent activation of caspase 3 (Figure 6C,D).

## 3. Discussion

Malignant Pleural Mesothelioma (MPM) is an aggressive tumor marginally impacted by standard chemotherapy regimens. Moreover, the lack of effective molecular therapies as well as the immune-evasive tumor microenvironment makes the treatment of MPM particularly challenging [5,6,7,8,9,10,42]. Because MPM currently lacks peculiar oncogenic drivers, we have explored the potential therapeutic efficacy of lurbinectedin, an alkylating agent which recently received FDA-conditional approval for the treatment of metastatic small cell lung cancer patients relapsing after chemotherapy [36].

We investigated the antitumor activity of lurbinectedin in a panel of 12 recently established primary MPM cell cultures. Our panel included all three MPM histotypes as well as cultures BAP1 positive and negative. Thus, although limited in terms of absolute number of cell lines, this panel is potentially representative of the different MPM phenotypes. Interestingly, we initially observed that lurbinectedin was effective at nanomolar concentrations and, as reported for other agents, its efficacy was independent of the BAP1 status. These data are particularly encouraging, although we are aware that freshly stabilized cultures could be potentially more sensitive to cytotoxic agents than what is usually observed at the clinical level. It is worthy of note, however, that three patients (UPN6, UPN10, UPN12) subsequently received trabectedin, a previous generation drug binding the minor groove of DNA, as second-line treatment. They did not show a superior clinical benefit compared to patients undergoing other treatments, indicating a limited efficacy of trabectedin. Interestingly, the MPM cells derived from these three patients had the highest IC_50_ to lurbinectedin. These data may suggest that the response obtained in our stabilized cultures is a good surrogate of the potential effect of drugs binding the DNA minor groove and targeting the DNA repair observed in vivo.

Our experiments revealed that, as a consequence of the intrinsic ability of lurbinectedin to bind the minor groove of DNA, the drug interferes with the cell cycle, delaying progression through the S-phase. Interestingly, MPM cells immediately responded to genotoxic stress as demonstrated by the phosphorylation of H2AX, an early marker of the cellular response triggered by DNA double-strand breaks. Moreover, we observed the activation of Chk1 and Chk2 as a direct consequence of the stalled replication induced by DNA damage, responsible for the accumulation of MPM cells in the S-phase of the cell cycle. Finally, in our setting, p53 stabilization was not associated with DNA repair but invariably resulted in a massive apoptotic response, as revealed by cleaved caspase 3 activity and irreversible DNA fragmentation detected by Comet assay.

Notably, the efficacy of lurbinectedin against MPM was maintained also upon long-term treatment, as assessed by both crystal violet viability and anchorage-independent growth assays, providing further evidence of its anticancer potential.

As a consequence of DNA damage, replication arrest, and induction of apoptosis, we propose that lurbinectedin impairs the tumorigenic potential of MPM cells, and our results provide support to the clinical data recently reported in a multicentric phase II trial in second- or third-line palliative therapy [37]. Speculatively, considering the high anti-proliferative effect, if the results of the present study will be confirmed in MPM PDXs, lurbinectedin could be potentially investigated in the front line setting, for instance for a short pre-operative treatment in the early stages of MPM. Indeed, the reduction of anchorage-independent growth ability suggests lurbinectedin as a potential cytoreductive agent that, if proven in animal models and at the clinical level, will allow more conservative/less invasive surgery. Finally, the efficacy in all histotypes, independently from the BAP1 status, confers to lurbinectedin a strong advantage compared to other drugs currently used in MPM treatment, since its use could be potentially considered for all patients.

## 4. Materials and Methods

### 4.1. Reagents and Chemicals

Cell culture plasticware was obtained from Falcon (Glendale, AZ, USA), Biofil (Indore, India), and Costar (Washingtone, DC, USA). Lurbinectedin (PM01183) was kindly provided by PharmaMar (Madrid, Spain).

### 4.2. Cells

Primary MPM cells were obtained from biopsies during explorative thoracoscopy or pleurectomy, performed at the Thoracic Surgery Division of AOU Città della Salute e della Scienza, Torino, Italy; AOU San Luigi Gonzaga, Orbassano, Italy, and AO of Alessandria, Biological Bank of Mesothelioma, Alessandria, Italy. Samples were anonymized by assigning an unknown patient number (UPN). Histological features of the original tumors and clinical features, including the first- and second-line treatment and the overall survival, of the corresponding patients are reported in Appendix A. Samples were minced in 1 mm^3^-pieces, enzymatically digested for 1 h at 37 °C with 0.2 mg/mL hyaluronidase and 1 mg/mL collagenase [5], centrifuged at 1200× *g* for 5 min and seeded at 1 × 10^6^ cells/mL density in DMEM advanced/F12 (Gibco, Dublin, Ireland) until passage #5, when cultures were shifted to DMEM/F12 nutrient mixture medium (Sigma, Saint Louis, MO, USA). All media were supplemented with 10% heat-inactivated fetal bovine serum (FBS) (Sigma), 1% L-glutamine, 1% penicillin/streptomycin. Cells reached a stabilization (i.e., rate of cell subculture ≤ 1/week) in 2 to 7 months. UPN#3, UPN#4, UPN#5, UPN#6, UPN#10, UPN#11, UPN #12 were directly put in culture. UPN#1, UPN#2, UPN#7, UPN#8, UPN#9 were established from patient-derived xenografts. All cell lines were cultured in a humidified incubator at 37 °C in 5% CO_2_ and routinely checked for Mycoplasma spp. contamination.

### 4.3. Patient-Derived Xenograft Generation

MPM patient-derived xenografts (PDXs) models were established from diagnostic tissue samples obtained at videothoracoscopy or during surgical pleurectomy. Each sample was implanted in the left or right side of the dorsal region of female NOD scid gamma (NSG) mice. A small piece of tumor was implanted subcutaneously and the wound was then stitched by surgical glue (Vetbond, Alcyon Italia, Cherasco, Italy). The tumor growth was monitored until the mass reached 2000 mm^3^. Then the animal was sacrificed by cervical dislocation, after anesthesia. The tumor area was shaved and disinfected with alcohol and the skin around the tumor was cut off. The tumor was divided into smaller pieces for re-implanting and collecting materials for further investigations. In the present work, the PDX platform was used as a tool to generate primary MPM cell cultures, stabilized in a shorter period (i.e., 2–3 months) than cells obtained directly from surgical procedures and used for pharmacological screening. To this aim, 0.2 g of tumors excised from the P1 generation of mice were digested to obtained a single-cell suspension [5] and put in culture as described in paragraph 4.2.

### 4.4. Immunohistochemical Analysis

The mesothelial features of cultures were confirmed by immunohistochemical (IHC) staining carried out on cells at passage 1. Specifically, cells were centrifuged at 1200× *g* for 5 min, fixed overnight in 4% *v*/*v* formalin at 4 °C, and then paraffin-embedded. The following antibodies were used: BAP-1 (Santa-Cruz Biotechnology, Santa Cruz, CA, USA, sc-28383, 1:100); Pan-cytokeratin AE1/AE3 (Dako, Agilent, Santa Clara, CA, USA, GA053, 1:500); Wilms Tumor-1 antigen (WT1) cl.6FH2 (Thermo Fisher Scientific, Waltham, MA, USA, MA1-46028, 1:10); Calretinin (Thermo Fisher Scientific, RB-9002-R7, 1:100). Mesothelial origin was confirmed if positivity for at least one between calretinin and WT1 was detected, as well as in the case of positivity for pancytokeratin. The histological features are reported in Table 1.

### 4.5. IC_50_ Calculation

Cells were seeded in 96-well plates at a density of 2 × 10^3^/well and serially diluted lurbinectedin (0.01 nM–100 nM) was added to the medium. After 72 h of treatment, IC_50_ was evaluated with CellTiter-Glo (Promega) according to the manufacturer’s instructions, using a Cytation 3 Imaging Reader (Bio-Tek Instruments, Winooski, VT, USA).

### 4.6. Crystal Violet Assay

For long-term proliferation, cells were seeded at a density of 4 × 10^3^/well in 12-well plates and treated with the indicated concentrations of lurbinectedin for 10 days. Subsequently, cells were fixed and stained with 5% *w*/*v* crystal violet solution in 66% *v*/*v* methanol and washed. Crystal violet was eluted by adding 10% acetic acid into each well. Quantification was performed by measuring the absorbance (570 nm) with Cytation 3 Imaging Reader (Bio-Tek Instruments).

### 4.7. Soft-Agar Assay

For anchorage-independent cell growth assay, cells were suspended in 0.45% type VII low-melting agarose in medium supplemented with 10% FBS at 1 × 10^5^ cells/well, plated on a layer of 0.9% agarose in 10% FBS medium in 6-well plates, and cultured for 20–30 days with the indicated concentrations of lurbinectedin.

### 4.8. Cell Cycle Analysis

Cells were plated at a density of 1.2 × 10^5^/well in 6-well plates and treated with the indicated concentrations of lurbinectedin for 24 h. Subsequently, cells were washed in PBS, treated with RNAse (167 μg/mL), and stained for 15 min at RT with propidium iodide (33 μg/mL). The cell-cycle distribution in G0/G1, S, and G2/M phases was analyzed by FACSCalibur flow cytometer (Becton Dickinson, Franklin Lanes, NJ, USA) and calculated using the CellQuest program (Becton Dickinson).

### 4.9. Apoptosis Detection Assay

MPM cells were plated at a density of 1.2 × 10^5^/well in 6-well plates and treated with the indicated concentrations of lurbinectedin for 72 h. Subsequently, floating and adherent cells were washed with PBS and stained with tetramethylrhodamine methylester perchlorate (TMRM) (200 nM) for 15 min at RT. The percentage of apoptosis was measured by FACSCalibur flow cytometer (Becton Dickinson) and calculated using the CellQuest program (Becton Dickinson).

### 4.10. Comet Assay

DNA damage was assessed by Single Cell Gel Electrophoresis assay (Comet assay) [43]. At least 100 nuclei were counted in each condition. The percentage of DNA in the tail was quantified using the CometScore software (TriTek Corp., Sumerduck, VA, USA).

### 4.11. Western Blot Analysis

Cells were washed with ice-cold PBS and incubated for 20 min on ice in 0.1% Triton X-100 lysis buffer (20 mM Tris HCl pH 7.4; 150 mM NaCl; 5 mM EDTA; 0.1% Triton X-100; 1 mM Phenylmethanesulfonyl fluoride; 10 mM NaF; 1 mM Na3VO4, supplemented with protease inhibitor cocktail). Cells were then centrifuged at 14,000× *g* for 15 min at 4 °C to remove any cellular debris. Protein lysates were subsequently quantified using DC protein assay (Bio-Rad), loaded in 4–12% NuPAGE Bis-Tris Protein Gels (Thermo Fisher Scientific) according to the manufacturer’s instructions, and transferred onto Hybond ECL nitrocellulose membranes. Blocking was performed with 5% Nonfat dried milk (PanReac AppliChem, Darmstadt, Germany) for 45 min at RT. Membranes were then incubated O/N at 4°C with the following antibodies: BAP-1 (Santa Cruz Biotechnology, sc-28383); phospho(Ser345) Chk1 (Cell Signaling, Danvers, MA, USA, 2348); phospho(Thr68) Chk2 (Cell Signaling, 2197); phospho(Ser15) p53 (Cell Signaling, 9286); GAPDH (Cell Signaling, 5174); cleaved Caspase3 (Cell Signaling, 9661); phospho(Ser139)-Histone H2A.X (Cell Signaling, 9718); rabbit IgG, HRP-linked (Cell Signaling, 7074); mouse IgG, HRP-linked (Cell Signaling, 7076). Proteins were detected with horseradish peroxidase-conjugated secondary antibodies and Pierce™ ECL Western Blotting Substrate.

### 4.12. Image Processing

Image acquisition was performed with Leica dmire2 microscope and with Olympus BX51. Images were processed with the ImageJ software package (https://imagej.nih.gov/ij/ accessed on 16 April 2021).

### 4.13. Statistical Analysis

All values were expressed as mean ± SEM and derived from at least two independent experiments. Statistical analyses were performed using Microsoft Excel and GraphPad Prism 5. Graphs were generated using Microsoft Excel and GraphPad Prism. Two-tailed Student’s t-test was used to evaluate statistical significance: ^NS^ *p* > 0.05; * *p* < 0.05; ** *p* < 0.01; *** *p* < 0.001; **** *p* < 0.0001.

## 5. Conclusions

Overall, our work proves the efficacy of lurbinectedin at nanomolar concentration against primary MPM cells. Although obtained in a relatively small cohort, that however is representative of the different MPM phenotypes, our results are particularly encouraging and put the basis for investigating lurbinectedin in different therapeutic settings of MPM.

## Figures and Tables

**Figure 1 cancers-13-02332-f001:**
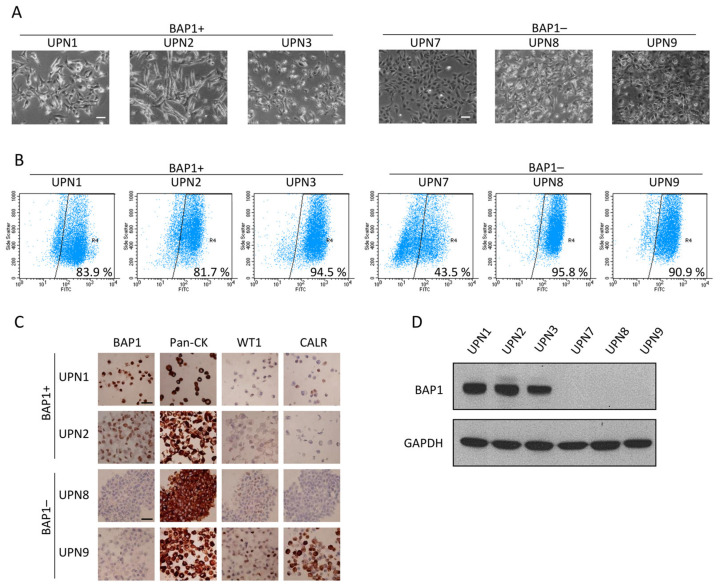
Characterization of patient-derived MPM cell lines. (**A**) Representative images showing different morphology of three BAP1 positive (BAP1+) and three BAP1 negative (BAP1−) MPM cell lines (scale bar = 100 µm). (**B**) Flow cytometry plot representing the percentage of pancytokeratine positive cells in the indicated MPM cell lines. (**C**) Immunohistochemical analysis of BAP1, pan-cytokeratin (pan-CK), Wilms tumor-1 antigen (WT1), and calretinin (CALR) in the indicated MPM cell lines (scale bar = 100 µm). (**D**) Western blot analysis showing BAP1 status of the reported MPM cell lines.

**Figure 2 cancers-13-02332-f002:**
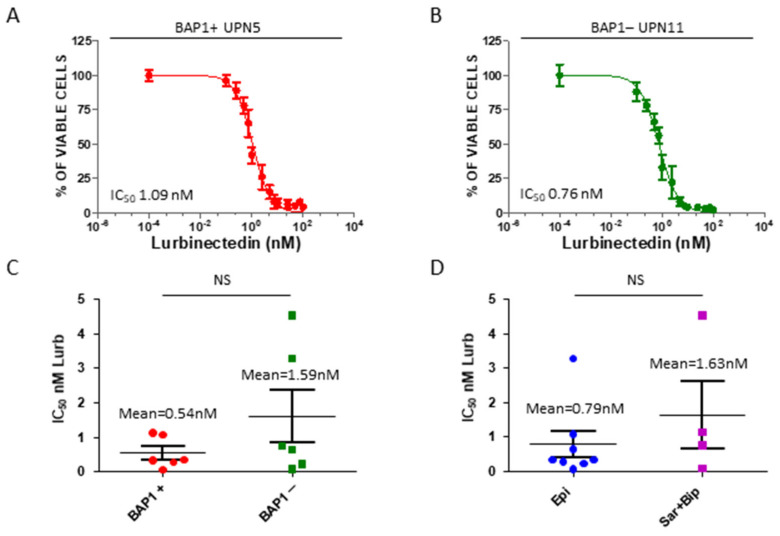
Patient-derived MPM cell lines sensitivity to lurbinectedin. (**A**,**B**) Representative dose-response curves and corresponding IC_50_ values of the two indicated MPM cell lines treated with lurbinectedin (0.1 nM–100 nM) for 72 h. (**C**) Dot plot of IC_50_ values measured in lurbinectedin-treated MPM cell lines positive or negative for BAP1 expression. ^NS^ *p* > 0.05. (**D**) Dot plot of IC_50_ values measured in lurbinectedin-treated MPM cell lines grouped according to the histological subtype. ^NS^ *p* > 0.05.

**Figure 3 cancers-13-02332-f003:**
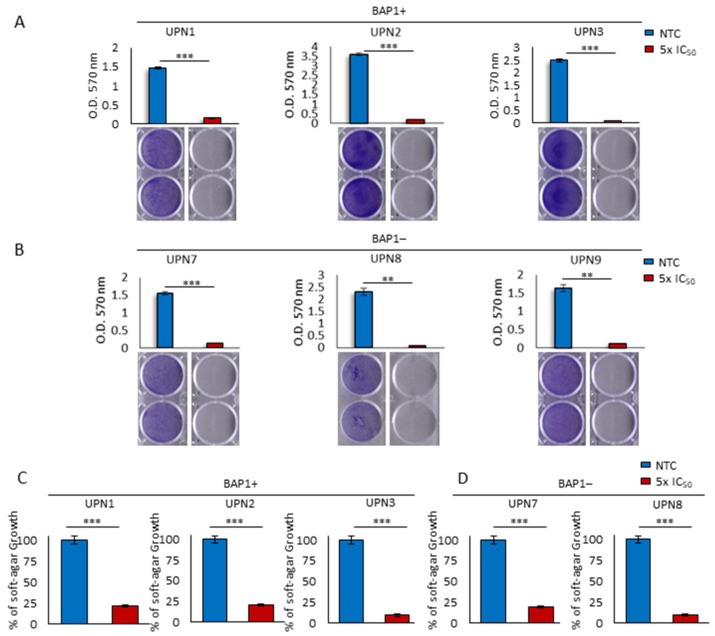
Lurbinectedin impairs long-term proliferation and anchorage-independent growth of MPM cell lines. (**A**,**B**) Representative pictures (lower panels) and quantification (upper panels) of crystal violet staining performed on the indicated MPM cell lines treated or not with lurbinectedin (5-fold the IC_50_) for 10 days. Data are expressed as means ± SEM; ** *p* < 0.01; *** *p* < 0.001. (**C**,**D**) Soft agar growth assay quantification of the indicated MPM cell lines treated or not with lurbinectedin (5-fold the IC_50_) for 20 days. The number of colonies obtained from untreated cells was set at 100%. Data are expressed as means ± SEM; *** *p* < 0.001.

**Figure 4 cancers-13-02332-f004:**
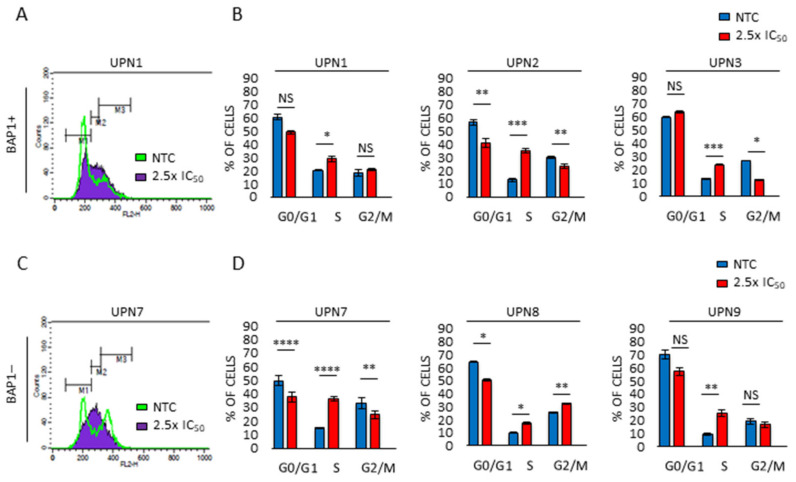
Lurbinectedin effects on cell cycle distribution. (**A**,**C**) Representative flow cytometry histogram showing the cell cycle distribution of the indicated MPM cell lines, treated (purple) or not (green) with lurbinectedin (2.5-fold the IC_50_) for 24 h. (**B**,**D**) Histograms displaying cell number percentage in each cell cycle phase (G0/G1, S and G2/M) of the indicated MPM cell lines, treated or not with lurbinectedin (2.5-fold the IC_50_) for 24 h. Data are expressed as means ± SEM; ^NS^ *p* > 0.05; * *p* < 0.05; ** *p* < 0.01; *** *p* < 0.001; **** *p* < 0.0001.

**Figure 5 cancers-13-02332-f005:**
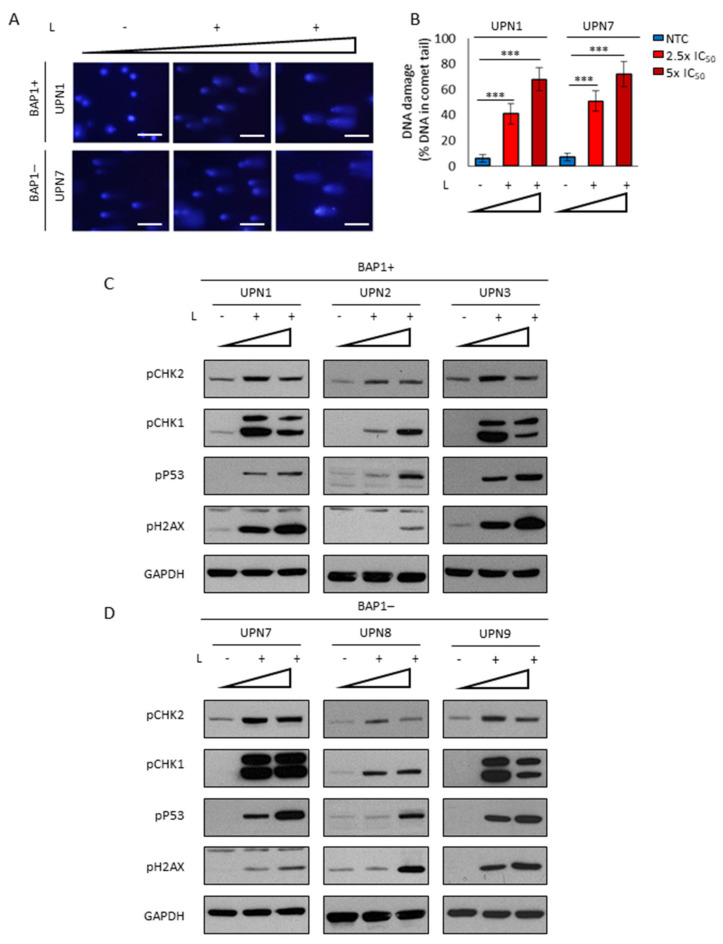
Lurbinectedin actively induces DNA damage response in MPM cell lines. (**A**) Representative Comet assay images of the indicated BAP1+ and BAP1− MPM cell lines treated or not with increasing lurbinectedin (L) concentrations (2.5-fold and 5-fold the IC_50_) for 24h (scale bar = 5 µm). (**B**) Histograms showing Comet assay data quantitation by CometScore software. Bars represent a percentage of total DNA in the tail. Data are expressed as means ± SEM; *** *p* < 0.001. (**C**,**D**) Western blot analysis for the indicated proteins in BAP1+ and BAP1- MPM cell lines treated or not with increasing lurbinectedin (L) concentrations (2.5-fold and 5-fold the IC_50_) for 24 h. GAPDH was used as a loading control.

**Figure 6 cancers-13-02332-f006:**
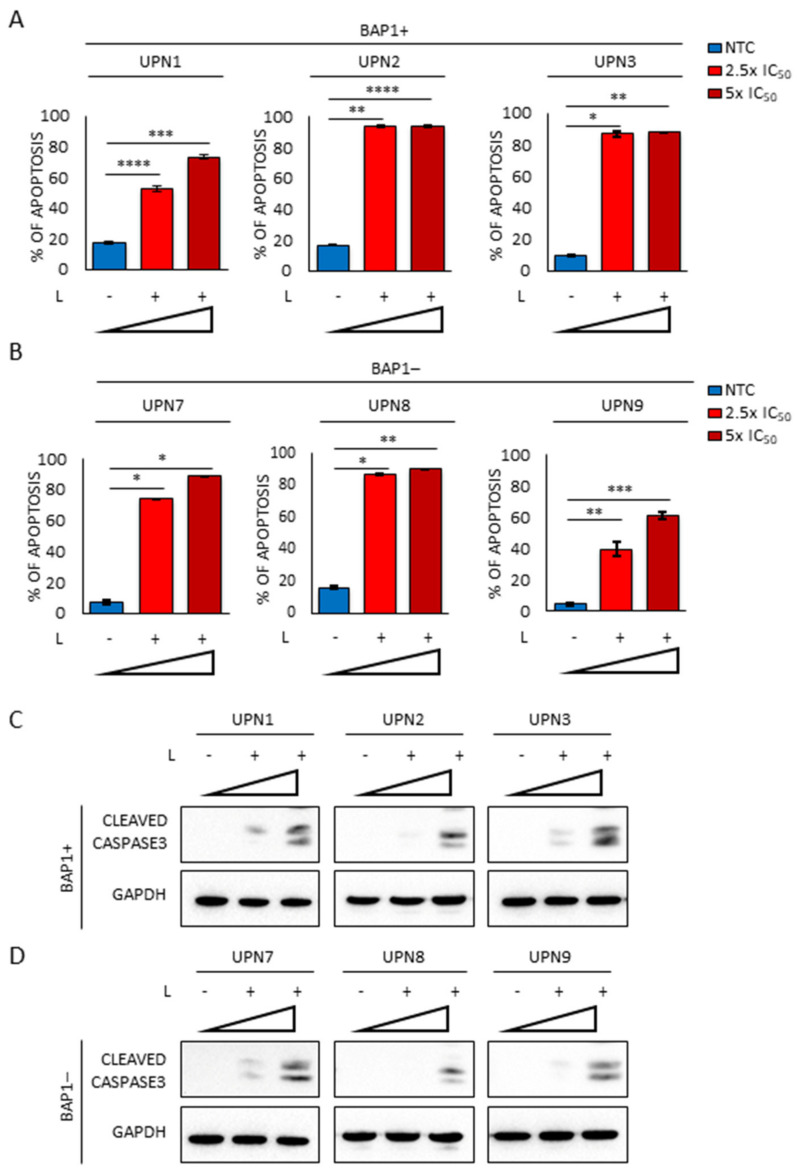
Lurbinectedin treatment strongly induces apoptosis in MPM cell lines. (**A**,**B**) Histograms representing the percentage of apoptotic MPM cells treated or not with increasing lurbinectedin (L) concentrations (2.5-fold and 5-fold the IC_50_) for 72 h. The apoptotic rate was measured by TMRM assay. Data are expressed as means ± SEM; * *p* < 0.05; ** *p* < 0.01; *** *p* < 0.001, **** *p* < 0.0001. (**C**,**D**) Western blot analysis of cleaved caspase 3 in MPM cell lines treated or not with increasing lurbinectedin (L) concentrations (2.5-fold and 5-fold the IC_50_) for 24 h. GAPDH was used as a loading control.

**Table 1 cancers-13-02332-t001:** Histological characterization of MPM cultures.

UPN	BAP1	Pan-CK	WT1	CALR
1	POS	POS	POS	POS
2	POS	POS	NEG	NEG
3	POS	POS	POS	NEG
4	POS	POS	POS	POS
5	POS	NEG	POS	POS
6	POS	POS	NEG	NEG
7	NEG	POS	POS	POS
8	NEG	POS	POS	NEG
9	NEG	POS	POS	POS
10	NEG	POS	POS	POS
11	NEG	POS	NEG	POS
12	NEG	POS	NEG	NEG

Results of the immunohistochemical stainings of MPM samples for BRCA1 associated protein-1 (BAP1), pancytokeratin (pan-CK), Wilms tumor-1 antigen (WT1), calretinin (CALR). POS: positive; NEG: negative.

**Table 2 cancers-13-02332-t002:** IC_50_ values of MPM cell lines treated with lurbinectedin.

UPN	IC_50_ L (nM)
1	0.073
2	0.33
3	0.28
4	0.35
5	1.09
6	1.13
7	0.085
8	0.65
9	0.23
10	3.29
11	0.76
12	4.54

## Data Availability

The data presented in this study are available in this article (and Appendix A).

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
