# Peer review of "Evaluation of the Preclinical Efficacy of Lurbinectedin in Malignant Pleural Mesothelioma"

_cancers, 2021, doi:10.3390/cancers13102332_

Round 1

Reviewer 1 Report

The manuscript entitled Evaluation of the preclinical efficacy of lurbinectedin in malignant pleural mesothelioma by Francesca Picca , Dario Pasquale Anobile , Paolo Bironzo , Marcello F. Lingua , Deborah Morena , Luisella Righi , Francesca Napoli , Mauro Giulio Papotti , Alessandra Pittaro , Federica Di Nicolantonio , Chiara Gigliotti , Federico Bussolino , Valentina Comunanza , Francesco Guerrera , Alberto Sandri , Francesco Leo , Roberta Libener , Pablo Aviles , Silvia Novello , Giorgio Vittorio Scagliotti * , Chiara Riganti * , Riccardo Taulli reports the evaluation of lurbinectedin for treatment of malignant pleural mesothelioma in vitro.

While the manuscript shows interesting findings, the authors could improve the quality of the manuscript by taking the aspects below into consideration:

Major aspects:

  • The authors claim the use of lurbinectedin would be beneficial for use in monotherapy for MPM patients, however, there is no in vivo data in the manuscript that support this. The authors should avoid far-fetched claims without supporting data.

Minor aspects:

* Have the authors tried applying 2.5x IC50 lurbinectedin at their crystal violet assays? That would be interesting to see in addition to their 5x IC50 data.

Author Response

Reply to Reviewer 1

Comments and Suggestions for Authors

The manuscript entitled Evaluation of the preclinical efficacy of lurbinectedin in malignant pleural mesothelioma by Francesca Picca , Dario Pasquale Anobile , Paolo Bironzo , Marcello F. Lingua , Deborah Morena , Luisella Righi , Francesca Napoli , Mauro Giulio Papotti , Alessandra Pittaro , Federica Di Nicolantonio , Chiara Gigliotti , Federico Bussolino , Valentina Comunanza , Francesco Guerrera , Alberto Sandri , Francesco Leo , Roberta Libener , Pablo Aviles , Silvia Novello , Giorgio Vittorio Scagliotti * , Chiara Riganti * , Riccardo Taulli reports the evaluation of lurbinectedin for treatment of malignant pleural mesothelioma in vitro.

While the manuscript shows interesting findings, the authors could improve the quality of the manuscript by taking the aspects below into consideration:

Major aspects:

The authors claim the use of lurbinectedin would be beneficial for use in monotherapy for MPM patients, however, there is no in vivo data in the manuscript that support this. The authors should avoid far-fetched claims without supporting data.

We agree that we need more in vivo data, currently ongoing in our research group, supporting our claim. We changed the text in the Simple Summary (line 53) and in the Discussion section (line 292).

Minor aspects:

Have the authors tried applying 2.5x IC50 lurbinectedin at their crystal violet assays? That would be interesting to see in addition to their 5x IC50 data.

We thank the Reviewer for the observation. We performed a series of pilot experiments on 6 MPM primary cell lines, representative of different histotypes and BAP1 mutational status, treated with lurbinectedin at 1.25X, 2.5X and 5X IC50. These experiments are the bases to set up combination treatments between lurbinectedin and other cytotoxic or immunological agents. Since we plan to include these data in a future dedicated work, we preferred not to show these data in the present manuscript, but to provide them confidentially to the Reviewer only (see “Figure for Reviewer 1”). Crystal violet assays showed a dose-dependent reduction of MPM cell viability (panel A-B). Notwithstanding the different sensitivity between the cell lines, lurbinectedin at 2.5X IC50 remained extremely effective against epithelioid and biphasic MPM cell lines, as well as in one sarcomatoid MPM cell line, but it lost its efficacy in the second sarcomatoid MPM cell line (panel C). However, since we evaluated only 3 epithelioid MPMs, 1 biphasic MPM and 2 sarcomatoid MPMs, the limited number of samples analyzed prevented us to draw any conclusive statement at this stage about the differential sensitivity between the histotypes to low doses of lurbinectedin. Also at 2.5X IC50 concentration, lurbinectedin seemed equally effective against BAP1+ and BAP1- cells (panel D), as at 5X IC50 concentration.

Reviewer 2 Report

This is a manuscript of basic research to evaluate the mechanism of action of lurbinectedin for malignant pleural mesothelioma. The manuscript is well-written, and the experiments are well-designed. Potentially, the data demonstrated in the manuscript could give important information to researchers or physicians in the area. However, before considering publication, a few revisions should be made.

  1. The authors describe that three patients received trabectedin treatment, and the MPM cells derived from those patients had the highest IC50 to lurbinectedin in discussion. These information looks important, so should be described in Materials section and Results section.
  2. Patient derived xenograft generation is described in Materials and Method section. But the results of xenograft model could not be found in results section.

Author Response

Reply to Reviewer 2

Comments and Suggestions for Authors

This is a manuscript of basic research to evaluate the mechanism of action of lurbinectedin for malignant pleural mesothelioma. The manuscript is well-written, and the experiments are well-designed. Potentially, the data demonstrated in the manuscript could give important information to researchers or physicians in the area. However, before considering publication, a few revisions should be made.

  1. The authors describe that three patients received trabectedin treatment, and the MPM cells derived from those patients had the highest IC50 to lurbinectedin in discussion. These information looks important, so should be described in Materials section and Results section.

As requested, we added the information about the three patients receiving trabectedin and the IC50 of the derived cell lines in the Results (line 163) and Materials and Methods (line 311) sections.

  1. Patient derived xenograft generation is described in Materials and Method section. But the results of xenograft model could not be found in results section.

In this work the patient derived xenografts (PDXs) were used only as a tool to generate primary MPM cells, stabilized in a shorter period than cells obtained directly from surgical procedures, to test the pharmacological potential of lurbinectedin. Describing the generation of the PDXs cohort, one of the first cohort in MPM research field, is beyond the aim of the present manuscript and will be included in a dedicated work. We provide this confidential information to the Reviewer only (see “Figure for Reviewer 2”).

We clarified the aim of using the PDX platform for the present study and the procedure to generate the primary cultures from PDXs in the Materials and Methods section (line 333).
